# Random Features for Normalization Layers

## Abstract

Can we can reduce the number of trainable parameters of neural networks by freezing a large portion of the initial weights? Training only BatchNorm parameters has shown great experimental promise, yet, the ability to express any potential target network would require a high amount of degrees of freedom. Even with sufficiently many parameters, contemporary optimization algorithms achieve only suboptimal performance. We systematically investigate both issues, expressiveness and trainability, and derive sparse random features which enjoy advantages in both aspects. In contrast to standard initialization approaches, they provably induce a well conditioned learning task and learning dynamics that are equivalent to the standard setting. They are also well aligned with target networks that can be approximated by random lottery tickets, which translates into a reduced bound on the number of required features. We obtain this bound by exploiting the layer-wise permutational invariance of target neurons, which applies to general feature distributions with good target alignment, and thus outline a path towards parameter efficient random features.

## 1 Introduction

Do we have to train all parameters of large neural network models or can we freeze a substantial portion at their random initial value? This question returns in different forms throughout the literature. Recent examples include (Rosenfeld & Tsotsos, 2019; Frankle et al., 2021; Williams et al., 2024). Training batch normalization (BN) or layer normalization parameters has shown to be particularly effective during finetuning of large language models (Ben Zaken et al., 2022; Lu et al., 2022; Kopiczko et al., 2024). It is standard to include them in the set of trainable parameters during finetuning and they are seldomly pruned during neural network sparsification. They are considered important, even though they usually provide technically redundant parameters that do not contribute to the expressiveness of neural networks, since they can be integrated in the weight parameters. It has been conjectured that BN parameters do not only provide benefits for the optimization procedure by facilitating large learning rates, preconditioning (Lange et al., 2022), and promoting flatness (Mueller et al., 2023), but also actively contribute to learning, as they find linear combinations of features (Giannou et al., 2023; Burkholz, 2024).

Affine transformations of features in general play a pivotal role in deep learning, as they are employed in most layer normalization techniques, pretrained model finetuning, or random feature models. Understanding the potential and limitations of linearly combining random features is central to the conjecture that we can reduce the number of trainable parameters of neural networks. (Frankle et al., 2021) found that training BN layers only while keeping the remaining parameters frozen to their initial value showed great promise in this regard. The hope is that this could reduce the overall memory footprint of models, as random values can be reconstructed from a random seed and only a small number of parameters needs to be saved. Subsequently, Giannou et al. (2023); Burkholz (2024) have rigorously proven that training the parameters of normalization layers (including BN) is equally expressive as training the full model, provided that the number of trainable parameters induces a sufficient number of degrees of freedom.

The ability to represent an arbitrary target therefore demands a number of available random features that scales quadratically in the number of nodes, which inflicts substantial costs in practice. Furthermore, training BN parameters from scratch with frozen random weights could so far not achieve competitive performance in comparison to the classic parameterization that also trains weights.

Frankle et al. (2021) posed therefore the question whether there could exist better suited random weight distributions that match to a smaller class of potential target networks.

Our answer is affirmative. By introducing maximally sparse, and optimally conditioned random features, we aim to address both open challenges, the trainability and high width requirement. By exploiting a permutational invariance of neural networks, we gain additional means to align a target network with given features, which is also a fundamental challenge in foundation model finetuning. The general question that we seek to answer is therefore this: How much target alignment is required and how much randomness or misalignment in the weights can we tolerate?

## 1.1 CONTRIBUTIONS

By investigating the question whether some random features are more effective in supporting the expressiveness and trainability of affine linear transformations, we make the following contributions:

1. We identify two mechanisms that distinguish the suitability of random feature distributions: a) how well they align with a target network and b) the general conditioning of the learning task that they induce.

2. Standard weight initializations induce random features, which are ill conditioned for larger learning problems, as we verify in experiments. This can be partially remedied by overparameterization, but higher degrees are required for larger systems.

3. Based on these insights, we propose two non-standard random feature distributions that are maximally sparse and enjoy ideal conditioning, which is reflected in their advantageous training dynamics which are equivalent to the standard setting.

4. Exploiting permutational invariances of target neural networks, we reduce the number of random features that are required for their reconstruction.

Our results, however, have a major caveat. While the random features that we propose solve the two open problems of trainability and sparsity, which are required to gain something from freezing a part of the neural network parameters, the features that we propose induce a parameterization that is equivalent to the original weight parameters of the neural network. This implies that training (a random subset of) the weight space instead of the original BN space is, in fact, the best we can do. This insight highlights the relevance of additional knowledge about the task either in form of inductive bias guiding the random weight distribution, domain knowledge, or task related features of a foundation model. Also in case of task aligned features, our theoretical analysis quantifies the value of conditioning and the consideration of permutation invariance.

## 1.2 RELATED WORK

**Benefits of Batch Normalization (BN)** Batch Normalization (BN) is one of the first and most prominent normalization techniques of neurons in neural networks (Ioffe & Szegedy, 2015). It introduces profound benefits for training speed and generalization, as it enables training with larger learning rates (Bjorck et al., 2018). It makes the training success robust to different choices of parameter initializations (De & Smith, 2020; Joudaki et al., 2023), and the loss landscape smoother (Santurkar et al., 2018). While BN defines effectively an affine linear transformation of a neuron's pre-activation, it also seems to simplify the learning task, as it tends to orthogonalize features (at least of linear neural networks) (Daneshmand et al., 2021) and also acts as preconditioner (Lange et al., 2022). Importantly, its conditioning operation applies to learning the weights of neural networks. In contrast, we study the conditioning of learning the parameters of the affine linear transformation on how different random feature distributions affect the learning task.

**Alternatives to BN** While highly effective, BN inflicts high computational costs and requires high amounts of memory, as it requires relatively high batch sizes to work well. Furthermore, it breaks the independence of minibatch samples and prevents adversarial training (Wang et al., 2022). To alleviate some of these disadvantages, multiple alternatives have been proposed, including weight normalization (Salimans & Kingma, 2016; Huang et al., 2017), weight standardization (Qiao et al., 2019), instance normalization (Ulyanov et al., 2016), instance enhancement batch normalization (Liang et al., 2020), or switch normalization (Luo et al., 2019). A combination of scaled weight

standardization and gradient clipping could even outperform BN (Brock et al., 2021). Our derivations for general affine transformations and different normalization mechanisms also cover these techniques.

**Parameter initialization for trainability** To reduce the need for BN, different initialization approaches that are tailored to specific architectures have been proposed (Balduzzi et al., 2017; Burkholz & Dubatovka, 2019; De & Smith, 2020; Zhang et al., 2018; Gadhikar & Burkholz, 2022). A particularly successful approach have been orthogonal weight initializations (Balduzzi et al., 2017; Burkholz & Dubatovka, 2019; Gadhikar & Burkholz, 2022) that generate dynamically isometric networks. Interestingly, we find that such orthogonal weight initializations induce suboptimal conditioning when affine linear transformations in two layers with random features approximate a standard target layer. In contrast, the random features that we derive enjoy ideal conditioning.

**Training only BN** If we commit to training only a small fraction of neural network parameters, experimental evidence suggests that BN parameters could be good candidates to be included in the set of training parameters. For instance, they have been found particularly effective in finetuning language models (Ben Zaken et al., 2022; Lu et al., 2022). Even if the neural network parameters are not pretrained but frozen to their initial values, training only the BN parameters was found to be more effective than training another random subset of neural network parameters while the remaining parameters were kept frozen (Frankle et al., 2021). This insights is supported by (Rosenfeld & Tsotsos, 2019), which, however, has focused on relatively short training periods. Combining both lines of research, VeRA is a finetuning technique that learns a perturbation $\Delta W$ of a pretrained weight matrix by adding linear combinations of random weight matrices (Kopiczko et al., 2024).

**Expressiveness of affine linear transformations** The experimental success of training only BN has been explained theoretically by (Giannou et al., 2023) and (Burkholz, 2024) for fully-connected and convolutional architectures, respectively. They derive bounds on the number of random features that are required for an affine linear transformation to approximate a general target layer. While these bounds suggest that a high number of features are required to obtain strong function approximators of general targets, we show that good feature and target alignment can reduce the number of required random features. Furthermore, we address an overlooked conditioning issue that hampers the experimental applicability of the derived insights.

**Random feature model** Our setting can be interpreted as a deeper version of the random feature model (Rahimi & Recht, 2007), where features are generated by a single random neural network layer to study fundamental effects of overparameterization and double descent (Belkin et al., 2019).

**Randomly masked neural networks** To show that there exist random features that have lower width requirements than the theoretically predicted results that guaranty the reconstruction of general targets, we utilize the literature on lottery tickets (Frankle & Carbin, 2019; Fischer & Burkholz, 2021a; Burkholz, 2022a;b). Randomly masked networks have shown surprisingly competitive performance on standard and large scale benchmark tasks (Su et al., 2020; Ma et al., 2021; Liu et al., 2021; Gadhikar & Burholz, 2024) up to sparsities of 90% and higher. This can be partially explained by width overparameterization (Gadhikar et al., 2023a), but also the existence of highly sparse representations of universal function approximators (Burkholz et al., 2022).

## 2 EXPRESSIVENESS OF AFFINE TRANSFORMATIONS

**Background and notation** Freezing weights and learning only the parameters of normalization layers can be reduced to a layerwise student-teacher setup, as visualized in Fig. 1 (a) & (b). To keep the exposition simple, we focus here on linear, fully-connected layers. Our results generalize to convolutional and residual architectures, which can consist of more layers with different activation functions (Burkholz, 2024). Each layer of a target network with weight parameters $\mathbf{W}^{(t)} \in \mathbb{R}^{c_2 \times c_0}$ is approximated by two student layers that keep their weights $\mathbf{W}^{(2)} \in \mathbb{R}^{c_2 \times c_1}$ and $\mathbf{W}^{(1)} \in \mathbb{R}^{c_1 \times c_0}$ frozen at their initial value and only adjust their normalization layer parameters $\gamma^{(2)}$ and $\gamma^{(1)}$ (Giannou et al., 2023; Burkholz, 2024). (For notational clarity, we omit the layer index $(l)$. Each layer is treated in the same manner.) The student thus has to minimize the loss

$$\mathcal{L}(\gamma^{(1)}, \gamma^{(2)}) = \sum_{ij} \left( \sum_k \gamma_i^{(2)} m_{(ij)k} \gamma_k^{(1)} - w_{ij}^{(t)} \right)^2, \tag{1}$$

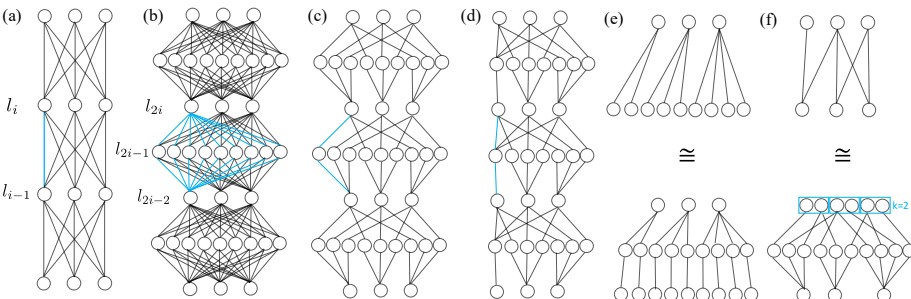

Figure 1: **Problem setup:** a) Target neural network. Each layer is approximated with two layers consisting of fixed (random) weights and trainable normalization layer parameters (b-d). The highlighted blue link is reconstructed utilizing the blue features in the other cases. (b) Most random features are dense. (c) Proposal of features (ID Cond) that induce a learning problem that is equivalent to (a). (d) A more parameter efficient variant of (c). (e) Target (top) and random features (bottom) for Thm. 4.2. (f) Target (top) and random features (bottom) for Thm. 4.3.

where $\mathbf{M} = (m_{(ij)k})$ is a $(c_0 c_2) \times c_1$-dimensional matrix with components $m_{(ij)k} = w_{ik}^{(2)} w_{kj}^{(1)}$. Note that details of different layer normalization approaches can be integrated into the parameters $\gamma^{(i)}$.

We have two options how we approach the reduction of this problem to linear regression. a) Eliminating $\gamma^{(2)}$ by replacing it as a function of $\gamma^{(1)}$ leads to the lowest established width requirement. b) Ignoring $\gamma^{(1)}$ (or setting all entries to 1), which leaves it free for utilization as normalization layer in practice, leaves the original structure of the matrix $\mathbf{M}$ intact. It is easier to analyze and usually better conditioned than a). However, it leads to a worse bound on the number of required random features $c_1$, since it does not utilize additionally available degrees of freedom. Our analysis generally applies to both scenarios and the conceptual differences are not significant.

**Width requirement for target reconstruction** In the following, we state the associated known results that provide a bound on $c_1$ to guarantee lossless reconstruction.

**Theorem 2.1** (Width requirement (Giannou et al., 2023; Burkholz, 2024))**.** *Problem (1) can be solved with $\mathcal{L}(\gamma^{(1)}, \gamma^{(2)}) = 0$ if the matrices $\mathbf{W}^{(1)}$ and $\mathbf{W}^{(2)}$ have full rank and $c_1 \geq c_2 c_0$. The parameters $\gamma_i^{(2)} = 1$ and $\gamma^{(1)} = \mathbf{M}^+ \mathbf{v}$ define a solution, where $\mathbf{M}^+$ denotes the Moore-Penrose inverse and $v_{(ij)} = w_{(ij)}^{(t)}$ is a flattened vector representation of the target. If we utilize $\gamma^{(2)}$ and $\gamma^{(1)}$ solves $\mathbf{M}\gamma^{(1)} = 0$ with $m_{(ij)k} = w_{i'k}^{(2)} w_{kj}^{(1)} - \frac{w_{ij}^{(t)}}{w_{ij_i}^{(t)}} w_{i'k}^{(2)} w_{kj_i}^{(1)}$, where $i'$ corresponds to a pivotal element of the target row, then $c_1 \geq c_2(c_0 - 1) + 1$ features suffice.*

This theorem restates results by (Giannou et al., 2023; Burkholz, 2024). The bound is far from encouraging, as the number of features $c_1$ scales quadratically with the number of target nodes. This is reasonable, since we need to attain the original number of degrees of freedom if we want to represent any possible target network. However, in practice, even this width requirement is not sufficient for standard random weight distributions, as the corresponding learning task tends to be badly conditioned, as we establish next.

## 2.1 BAD CONDITIONING IMPLIES HIGHER WIDTH REQUIREMENT IN PRACTICE

Training only BN parameters from scratch with frozen random weights could generally not recover the performance of a model that was trained with the original parameterization (Frankle et al., 2021) despite the theoretical existence of a solution (Burkholz, 2024). Training in a student-teacher setup using a target layer as a teacher could provide empirical evidence that it should be possible to train exclusively the normalization layers, but also this reconstruction performed well only with more features than the theoretical bound requests. The following section identifies the main source of the issue, namely, that the learning task is ill conditioned, as Fig. 2 (b) illustrates. Fig. 2 (c)&(d) furthermore explores the impact of different mitigation strategies that pre-condition the learning task.

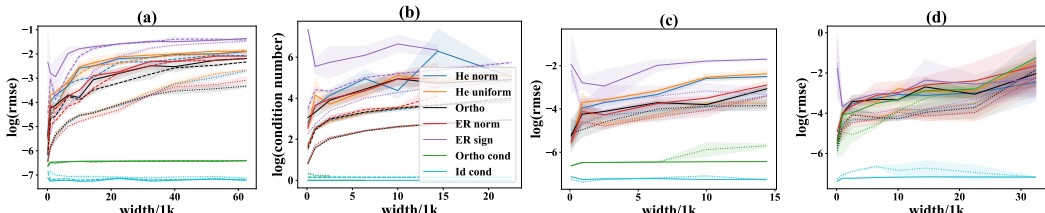

Figure 2: **Conditioning:** (a) $\text{Log}_{10}$ of the root mean squared error (rmse) obtained by LBFGS reconstructing random targets and random features following the same distribution. Solid lines: The exact number of features that are required for perfect reconstruction are provided. Dashed line: 10 additional features. Dotted line: 100 additional features. (b) Condition number of the respective random features. It could not be identified due to numerical instabilities if it is not shown. (c) As in (a) but $\mathbf{M}$ is first preconditioned with a SVD to improve LBFGS. (d) As (c), but we pre-condition, then utilize the Pytorch linear algebra solver *torch.linalg.solve* ane finetune with LBFGS.

While they reduce the approximation error, their effect is limited and they do not scale to realistic neural network sizes. The spectrum of the random feature matrix $\mathbf{M}$ is the core of the problem.

**Standard random features** While the matrices $\mathbf{W}^{(1)}$ and $\mathbf{W}^{(2)}$ would be pre-trained in finetuning tasks, here, we aim to analyze how far we can get with random features that only encode little information about the potential target. (Frankle et al., 2021; Burkholz, 2024) had primarily investigated freezing the weights to their initial value and therefore considered the popular He (He et al., 2015) and orthogonal (Hu et al., 2020) initializations so that $\mathbf{W}^{(1)}$ and $\mathbf{W}^{(2)}$ are independent. In addition, we also study sparse variants that are randomly masked (with probability $1/2$) and thus have random iid normally or binary $w_{ij} \in \{-1, 1\}$ distributed entries, which are labeled ER norm or ER sign.

**Spectral properties** The singular values $S$ of the appropriately scaled feature matrix $\mathbf{M}$ are a crucial factor in determining the hardness of the optimization problem that we try to solve, which is commonly measured by the condition number $\kappa(\mathbf{M}) = \overline{s}/\underline{s}$, i.e. the quotient of the maximum singular value $\overline{s}$ and the minimum nonzero singular value $\underline{s}$. The singular values follow approximately the Wigner's semi-circle law (see Fig. 3), as the following theorem states.

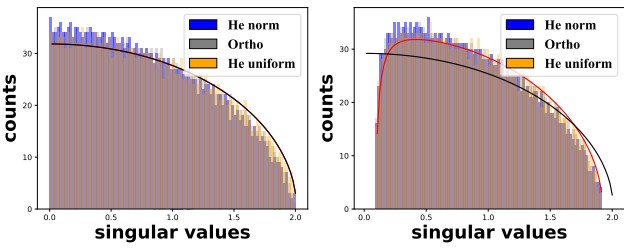

Figure 3: **Spectrum:** Histograms of the singular values of a random matrix $M$ with dimensions $c_0 = c_2 = 50$ and $c_1 = 2500$ (left) or $c_1 = 3000$ (right). The black line marks the semi-circle law with scale $\lambda = 1$ and the red one with scale $\lambda = (c_0 c_2)/c_1$, which adjusts for the excess features.

**Theorem 2.2** (Spectral Density). *The singular values of the matrix* $\mathbf{M} = (m_{(ij)k})$ *with* $m_{(ij)k} = w_{ik}^{(2)} w_{kj}^{(1)}$, *where the factors* $w_{ik}^{(2)}$, $w_{kj}^{(1)}$ *are independent, have uniformly bounded fourth moments, and the variance* $\text{Var}(m_{(ij)k}) = 1/(c_0 c_2)$, *are asymptotically (for* $c_0 c_2 \to \infty$*) distributed with probability density* $p(x) = \frac{1}{\lambda \pi x} \sqrt{(x^2 - \lambda_-^2)(\lambda_+^2 - x^2)}$, *where* $\lambda = \max((c_2 c_0)/c_1, c_1/(c_2 c_0))$, $\lambda_- = 1 - \sqrt{\lambda}$, *and* $\lambda_+^2 = 1 + \sqrt{\lambda}$ *and* $x \in [\lambda_-, \lambda_+]$.

The proof is given in the appendix. Note that the entries of $\mathbf{M}$ are not all independent but they are uncorrelated and the columns are independent, which we can exploit to show that the eigenvalues of $\mathbf{M}\mathbf{M}^T$ follow a Marchenko–Pastur distribution (Bryson et al., 2021).

**Double Descent** The biggest problem occurs exactly at the point of the minimum width requirement, where the linear system of equations becomes solvable, i.e., the interpolation point, where $\lambda = 1$ so that $\lambda_-^2 = 0$. The minimum singular value can get arbitrarily close to 0, which induces a high condition number. Learning components in corresponding singular directions becomes hard. On average, the corresponding condition number further increases for larger systems, which is associated

with more erroneous target reconstruction (see Fig. 2). Yet, it goes through a double descent (Poggio et al., 2020), which means that it peaks at the interpolation threshold. More or fewer features (s.t. $\lambda > 1$) make the problem algorithmically better solveable. However, to fight problematic conditioning, the number of excess or missing features must scale with the target dimension $c_0 c_2$, which inflicts potentially high computational and memory costs or hampers the reconstruction. Our declared goal is to reduce the number of required features by taking the properties of the target into account. Yet, leaving out too many (random) features still reduces the expressive power of a layer and thus limit its capacity. For that reason, we need additional ways to deal with bad conditioning. The random features, which we propose, enjoy perfect conditioning by design.

**Orthogonal weights are not well conditioned.** Note that orthogonal weights are considered to be an excellent choice for neural network initialization, because each matrix $W$ separately has perfect condition number 1 (and thus contributes to dynamical isometry (Balduzzi et al., 2017; Burkholz & Dubatovka, 2019; Gadhikar & Burkholz, 2022; Nowak et al., 2024)). Perhaps surprisingly, this does not transfer to $\mathbf{M}$ and thus to task of learning normalization parameters. While the columns of $\mathbf{M}$ are orthogonal, the rows are not, as $\sum_k w_{ik}^{(2)} w_{kj}^{(1)} w_{i'k}^{(2)} w_{kj'}^{(1)} \neq 0$.

# 3 ADVANTAGEOUS RANDOM FEATURES

One of our main contributions is the derivation of random features that are specifically designed to support learning normalization layers, as they are perfectly conditioned and provably induce well behaved learning dynamics.

**ID cond features** The ID cond features are visualized in Fig. 1 (c) & (d). Let us first assume that $\mathbf{M}$ has width $c_1 = c_0 c_2$. Every feature $k$ can then be associated with a target dimension $(i, j)$ so that

$$m_{(ij)k} := 1, \text{ if } k = k_{ij}, \text{ and } m_{(ij)k} := 0 \text{ otherwise.} \tag{2}$$

Without loss of generality, we assume that the indicees $k$ are arranged so that the feature matrix becomes the identity matrix $\mathbf{M} = \mathbf{I}$. To achieve the minimum width requirement, however, we have to remove $c_2 - 1$ features as illustrated in Fig. 1 (d) and utilize $\gamma^{(2)}$. For this case, we define:

$$m_{(ij)k} := 1, \text{ if } k = k_{ij}, \text{ and } m_{(ij)k} := 0 \text{ otherwise for } j > 1, \ m_{(i1)k} := 1 \text{ for } k = k_1. \tag{3}$$

This matrix cannot be perfectly conditioned, but the solution of the target reconstruction is still straight forward: $w_{ij}^t = \gamma_{k_i}^{(2)} \gamma_{k_{ij}}^{(1)}$ for $j > 1$ and $w_{i1}^t = \gamma_{k_i}^{(2)} \gamma_{k_1}^{(1)}$. Note that $\gamma_{k_1}^{(1)}$ is a free parameter that results from the scale invariance of consecutive neural network layers.

**Ortho cond features** Following a similar principle, i.e., disentangling orthogonal dimensions, we can also define more general orthogonal features, where $m_{(ij)k} = u_{iq}^{(2)} u_{dj}^{(1)}$ with $k = (qd)$. $\mathbf{U}^{(2)}$ and $\mathbf{U}^{(1)}$ are random, quadratic orthogonal matrices. ID cond can be seen as special case where the orthogonal matrices are identity matrices. In our experiments, we draw random Ortho cond features by sampling $\mathbf{U}^{(1)} \sim O(c_0)$, $\mathbf{U}^{(2)} \sim O(c_2)$ independently from the respective orthogonal ensemble.

**Ortho cond features are perfectly conditioned.** Our proposed features are perfectly conditioned by design, since the corresponding feature matrix $\mathbf{M}$ is orthogonal. This is easy to see, since the rows inherit the orthogonality from their components so that $\sum_{(ij),(qd)} m_{(i'j')(qd)} m_{(ij)(qd)} = \sum_q u_{iq}^{(2)} u_{i'q}^{(2)} \sum_d u_{dj}^{(1)} u_{dj'}^{(1)} = \delta_{ii'} \delta_{jj'}$, where $\delta$ denotes the Kronecker delta. This also holds for ID cond features, which are a special, maximally sparse case.

Ortho cond features in general are ideal for the learning task to reconstruct targets layerwise in a student-teacher setting. In practice though, we would want to train all layers simultaneously. As it turns out, ID cond features overcome also the challenge of ill-conditioned training from scratch, as we can achieve the same performance as training in the original weight space.

**Theorem 3.1** (Equivalent Features). *Learning with ID cond features (2) is equivalent to learning a neural network in its original parameterization with $w_{ij}^{(t)} = \gamma_{k_{ij}}^{(1)}$. The corresponding feature matrix $\mathbf{M} = \mathbf{I}$ is maximally sparse and perfectly conditioned.*

The proof is given in the appendix. ID cond features induce not only advantages for optimization, they are also maximally sparse, which makes them attractive from a computational point of view. Yet,

so far we have nothing gained compared to training just the original weights. Next, we investigate under which conditions we can reduce the theoretical width requirement.

# 4 PERMUTATION INVARIANCE

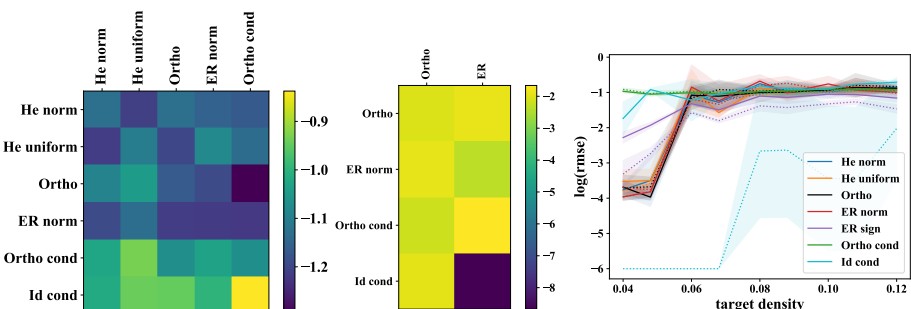

Figure 4: **Alignment after permutation with low rank targets:** The target matrix is drawn randomly as $\mathbf{W}_2^{(t)}\mathbf{W}_1^{(t)}$, where each $\mathbf{W}_i$ follows the column distribution. The target dimensions are $(c_0, c_2) = (50, 50)$ with hidden dimension $c_1^{(t)}$. Similarly, the matrices comprising the feature matrix $\mathbf{M}$ follow the row distribution, where the number of random features $c_1$ is lower than the width requirement. The average reconstruction error (over 10 independent samples) after Greedily permuting the target is color coded. Left: $c_1^{(t)} = 100$, $c_1 = 2440$. Middle: $c_1^{(t)} = 300$, $c_1 = 2400$, Right: The Bernoulli $Ber(p)$ iid mask, where $p$ is the expected target density. $c_1 = 2400$. Solid lines: Reconstruction error without permutations. Dotted line: Permuting target neurons for better match with random features. Mean and standard errors are computed for 10 seeds.

As it turns out, the singular value space of the feature matrix $\mathbf{M}$ determines not only the numerical difficulty of our optimization problem of interest, it also is key to answering the question whether we can further reduce the theoretical width requirement by exploiting the so far disregarded permutation invariance of a target. As implication, our target reconstruction problem has another degree of freedom that we can optimize, i.e. permutation matrices $\mathbf{P}$ that permute the rows of the target matrix.

$$\min \mathcal{L}(\gamma^{(1)}, \gamma^{(2)}, \mathbf{P}), \text{ where } \mathcal{L}(\gamma^{(1)}, \gamma^{(2)}, \mathbf{P}) = \left\|\mathbf{M}(\mathbf{P})\gamma^{(1)} - vec(\mathbf{P}\mathbf{W}^{(t)})\right\|_2^2. \quad (4)$$

In the following, we study the opportunities resulting from optimizing $\mathbf{P}$ in a target specific manner. Our guiding question will be whether this can reduce the width requirement on $\mathbf{M}$.

**Can we leverage properties of the target?** The next theorem derives a criterion that we can optimize to find permutations that minimize the mean squared error with respect to the target if we have only $r$ features. Based on this result, we have defined a Greedy algorithm (see appendix) that seeks to maximize the alignment of features and targets in Fig. 4.

**Theorem 4.1** (Target Alignment). *Assume that $\mathbf{M}$ has rank $r$ and singular value decomposition $\mathbf{M} = \mathbf{U}\mathbf{S}\mathbf{V}^T$ with singular values sorted in decreasing order. Let $\mathbf{P}$ be a permutation of the rows of the target $\mathbf{W}^{(t)}$ and $\mathbf{v}^{(t)}(\mathbf{P}) = vec(\mathbf{P}\mathbf{W}^{(t)})$ be the flattened vector representation of the permuted target. Then, the approximation error is minimized by $\min_{\mathbf{P}} \min_{\gamma} \left\|\mathbf{M}\hat{\gamma} - \mathbf{v}^{(t)}(\mathbf{P})\right\|^2 = \min_{\mathbf{P}} \left\|\mathbf{P}_r\mathbf{U}^T\mathbf{v}^{(t)}(\mathbf{P})\right\|^2$, where $\mathbf{P}_r = \begin{bmatrix} \mathbf{0} & \mathbf{0} \\ \mathbf{0} & \mathbf{I}_r \end{bmatrix}$ projects a vector to its last $r$ components.*

The proof is given in the appendix. It seems intuitive that the permutation seeks to align the target with the leading singular vectors of the features $\mathbf{M}$ and thus minimize its variation in the directions that cannot be covered by linear combination of the columns of $\mathbf{M}$. Accordingly, the alignment between the target and features $\mathbf{M}$ in the singular value space defines the critical measure of the remaining approximation error $\left\|\mathbf{P}_r\mathbf{U}^T\mathbf{v}^{(t)}\right\|^2$.

**Target alignment** Note that the above statement applies to any feature matrix $\mathbf{M}$. It does not assume that $\mathbf{M}$ is random but also applies to pre-trained features. Assuming such high alignment lies at the

heart of many fine-tuning techniques. The expressiveness of the approach can be increased by also updating the singular vectors, which can aid generalization if the alignment of the pre-trained weights and the target weights is not sufficient (Lingam et al., 2024).

**Alignment with random targets**   If our target has a low rank or is sparse, matching should be easier and reduce our width requirement. As the reconstruction success hinges on target and feature alignment, it could help if a random target, though independent, follows the same distribution as the features and thus shares their inductive bias. Yet, Fig. 4 suggests that permutations obtained by our Greedy matching algorithm hardly reduce our width requirement. Only highly sparse targets can be reconstructed with fewer sparse ID cond features. However, ID cond features seem less suitable than other random features to exploit low rankness of dense targets. Apart from sparsity, we do not observe strong positive associations between similar target and feature distributions. The standard dense features follow approximately Wigner's semicircle law of Gaussian matrices. Thus, they all align with similarly distributed targets. Visually, they are distinguishable from the two better conditioned feature distributions, corroborating our theoretical insights.

**Sparse targets** Sparse targets and features, however, present more promising candidates to realize width reductions. Fig. 4 (right) shows that permutations that improve the target alignment (see appendix for matching algorithm) can make reconstruction possible at a feature width $c_1$ that is smaller than our previous bound. However, the target density at which permutations help effectively is disappointingly low. Can we understand why this is the case?

**Theorem 4.2** (Matching sparse targets and features). *Let the target matrix $\mathbf{W}^{(t)}$ be a sparse matrix with in-degrees $d_i = \sum_j \delta_{w_{ij}^{(t)} \neq 0}$ and the feature matrix $\mathbf{M}$ correspond to a random $Ber(p)$ matrix (i.e. sparse Id cond features). Then, the probability that the (permuted) target can be accurately matched is upper bounded by $\mathbb{P}(\mathbf{W}^{(t)} \subset \mathbf{M}) \leq \Pi_i \left(1 - \left(1 - p^{d_i}\right)^{c_2}\right)$.*

*Proof Outline.* The proof is provided in the appendix. The main idea is to consider a special case that attains the lower bound. If all target output neurons are connected to mutually exclusive sets of input neurons (see Fig. 1 (e)), then all student outputs are potential matches for a target neuron.

This result explains, why permutations can only improve reconstruction performance at relatively low target density and high feature density (see Fig. 4). To see this, let us assume for simplicity that all target neurons have the same degree $d_i = d$. Theorem 4.2 implies that we would need at least a density of order $1 - (\delta/c_2)^{1/c_2}$ to represent a target with $dc_2$ edges with success probability $1 - \delta$. For reference, a target with $c_2 = c_0 = 50$ and $d = 10$ (and thus density 0.2) would require more than $p = 0.8$ for target reconstruction with probability $1 - \delta > 0.8$. Our target would therefore have to be very sparse (i.e. have small $d$) to allow for a significant reduction in the feature width (i.e. small $c_1$ or $p$). This conclusion is in line with Fig. 4 (right), where maximizing the target alignment by permuting output neurons affects the reconstruction loss positively only when the target has a low density.

**Could we still succeed with fewer random features?** Theorem 4.2 has highlighted a considerable limitation of learning based on sparse random features (i.e. small density $p$). Random directions in high-dimensional spaces are unlikely to align with unknown targets. And, yet, the unreasonable effectiveness of random pruning calls this into question (Liu et al., 2021; Gadhikar et al., 2023a). Related lottery ticket existence proofs that try to explain this fact postulate the existence of much sparser targets of lower width (Fischer & Burkholz, 2021a; da Cunha et al., 2022; Burkholz, 2022a;b) and smaller number of layers (Burkholz, 2022b; Fischer & Burkholz, 2021b). Making $c_2$ wider than the target, would indeed provide many more permutation options, as we prove next.

**Theorem 4.3** (Lower dimensional target). *A random target with iid $Ber(q)$ entries and $c_2^{(t)}$ output neurons can be perfectly matched with probability $1 - \delta$ with random ID cond features with expected density $p$ if $c_2 = kc_2^{(t)}$ with $k \geq \frac{\log\left(1 - (1-\delta)^{1/c_2^{(t)}}\right)}{\log(1 - (1 - q(1-p))^{c_0})}$.*

The proof is given in the appendix and the setup is visualized in Fig. 1 (f). Note that the special target in Fig. 1 (e) would only need $k \approx (1 - \delta)^{1/c_2}/(c_2 p^d)$. In practice, (Frankle et al., 2021) found that approximately $1/3$ of activations are switched off when only BN parameters are learned, suggesting a $k \approx 1.5$. In summary, sparse targets can be matched under certain conditions, which is corroborated by empirical evidence that random masks work in practice (Liu et al., 2021; Gadhikar & Burholz, 2024) and ID cond features are provably trainable from scratch.

***Conclusion.*** *If a task is solvable by training an iid randomly masked neural network with density $p_l$, then $c_1^l = p_l^l c_0^l c_2^l$ random ID cond features can achieve the same generalization performance.*

In practice, $p_l^l \approx 0.2$ is often feasible without significant performance loss. This concludes our theory, as we have established that there exist random sparse features that support learning only normalization layers and these features are perfectly conditioned.

## 5 EXPERIMENTS

Our experiments have identified bad conditioning as major obstacle in approximating given targets in a student teacher setting (see Fig. 2). Mitigating this issue by preconditioning does not scale to typical neural network sizes and is not as effective as using our well conditioned features ID cond and Ortho cond. Furthermore, we have permuted targets to align with our features to improve the reconstruction error

| cond. num. | 1 | 5413 | 10123 | 1541601 |
|---|---|---|---|---|
| train acc | 81.89 | 79.06 | 79.1 | 76.28 |
| test acc | 76.89 | 75.84 | 75.6 | 73.14 |

Table 1: ResNet50 trained on ImageNet with ID features scaled by a spectrum that was drawn from a semi-circle law with the reported average condition number assuming $c_0 c_2 = 10^6$. Note that many ResNet50 layers are larger are therefore worse conditioned.

(see Fig. 4) based on Thm. 4.2, which solidifies our theoretical insight that permutations make a relevant difference, yet, primarily for matching extremely sparse targets. In addition, we next explore the implications of conditioning for training from scratch.

**CIFAR10** To resemble our theoretical setup, we replace the convolutional and fully connected layers of a Vgg18 with width 100 by two linear random He normally distributed weight layers for which we train linear combinations of the intermediary neurons on CIFAR10. If we train the network with intermediary width $c_1 = 9 * 100 * 100 + 100$ with 100 excesss features (without warmup) for 300 epochs, we obtain an accuracy of $(86.64 \pm 0.08)\%$ (averaged over 3 independent runs). Training longer for 500 epochs with warmup to combat problematic conditioning increases the accuracy $(90.08 \pm 0.03)\%$. Despite using excess features, it cannot compete with training the network in the original parameterization, which can achieve an accuracy in the order of $94\%$. These experiments are limited to small scale settings, as the required width of the middle random feature layer is substantial.

**ImageNet** To study the effects of bad conditioning in larger scale settings, where they should be even more detrimental, we study rescaled ID precond features with arbitrary singular values $s_{(ij)}$, which correspond to a parameterization $w_{ij} s_{ij}$ where $s_{ij}$ is a fixed scaling and $w_{ij}$ is trainable. Table 1 reports the results for spectra that resemble the classic random distributions (which approximately follow the semi-circle law). As expected, ill conditioning affects the training outcome negatively. The original parameterization corresponds to the ideally conditioned case.

## 6 DISCUSSION

We have affirmatively addressed the open question posed by Frankle et al. (2021) regarding the existence of random weight distributions that are more conducive than others to training only normalization layers, while keeping all other parameters fixed. While standard random weight initializations often suffer from poor conditioning, artificially increasing the number of features—though impractical—can improve the optimization landscape. Conversely, reducing features also improves conditioning but at the cost of significantly reducing model expressiveness. To overcome these obstacles, we have introduced random weight distributions that are perfectly conditioned, maximally sparse, and provably trainable from scratch. They can also lower the theoretical width requirement. However, they are mathematically equivalent to the original parameterization, implying that training normalization layers alone with random weights (Frankle et al., 2021; Burkholz, 2024) does not yield performance gains over standard training. Nevertheless, our analysis suggests that fine-tuning normalization layers in pre-trained foundation models can enable efficient learning when the target task aligns well with the model's singular value structure. These findings provide fundamental insights into how the structure of weight matrices influences the effectiveness of normalization-layer training, and could inform future approaches to jointly optimizing weights and normalization parameters.

## REPRODUCIBILITY STATEMENT

All theoretical statements are explained and proven in the appendix. The code to reproduce the experiments is provided as supplement.

## LLM STATEMENT

To improve fluency, large language models have been used to polish the discussion section.

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

# A APPENDIX

## A.1 PROOFS OF THEOREMS

## A.2 PROOF OF THEOREM 2.2

*Statement* (Spectral Density, Thm. 2.2). The singular values of the matrix $\mathbf{M} = (m_{(ij)k})$ with $m_{(ij)k} = w_{ik}^{(2)} w_{kj}^{(1)}$, where the factors $w_{ik}^{(2)}$, $w_{kj}^{(1)}$ are independent, have uniformly bounded fourth moments, and the variance $\mathrm{Var}(m_{(ij)k}) = 1/(c_0 c_2)$, are asymptotically (for $c_0 c_2 \to \infty$) distributed with probability density

$$p(x) = \frac{1}{\lambda \pi x} \sqrt{(x^2 - \lambda_-^2)(\lambda_+^2 - x^2)},$$

where $\lambda = \max((c_2 c_0)/c_1, c_1/(c_2 c_0))$, $\lambda_- = 1 - \sqrt{\lambda}$, and $\lambda_+^2 = 1 + \sqrt{\lambda}$ and $x \in [\lambda_-, \lambda_+]$.

*Proof.* This theorem would follow trivially if the entries $m_{(ij)k} = w_{ik}^{(2)} w_{kj}^{(1)}$ of $\mathbf{M}$ were independent. However, note that $m_{(ij)k}$ and $m_{(ij')k}$ as well as $m_{(i'j)k}$ and $m_{(i'j)k}$ are dependent because they share a factor. They are still uncorrelated, as we still have $\mathbb{E}(m_{(ij)k} m_{(i'j)k}) = \mathbb{E}\left(w_{ik}^{(2)} w_{i'k}^{(2)} (w_{kj}^{(1)})^2\right) = \mathbb{E}\mathbb{E}\left(w_{ik}^{(2)}\right) \mathbb{E}\left(w_{i'k}^{(2)}\right) \mathbb{E}\left(w_{kj}^{(1)}\right)^2 = 0$, because the weights are centered. Furthermore, different columns $k$ and $k'$ are independent. Accordingly, the distribution of $\mathbf{M}$ follows the block-independent model by Bryson et al. (2021). The size of a block of dependent entries is $c_2$ and therefore $c_2/(c_2 c_0) = 1/c_0 \to 0$ as we increase the size of $\mathbf{M}$ so that $c_0 \to \infty$. According to Theorem 1.3 by Bryson et al. (2021), the spectral distribution of $\mathbf{M}^T \mathbf{M}$ converges with probability 1 weakly in distribution to the Marcenko-Pastur law with parameter $\lambda$. $\qquad\square$

## A.2.1 PROOF OF THEOREM 3.1

*Statement* (Equivalent Features, Thm. 3.1). Learning with ID cond features (2) is equivalent to learning a neural network in its original parameterization with $w_{ij}^{(t)} = \gamma_{k_{ij}}^{(1)}$. The corresponding feature matrix $\mathbf{M} = \mathbf{I}$ is maximally sparse and perfectly conditioned.

*Proof.* Our first objective is to show that ID cond features lead to an equivalent parameterization $w_{ij}^{(t)} = \gamma_{k_{ij}}^{(1)}$. This follows directly from the construction, where we have to track the indices from the operation that flattens matrices to vectors. For instance, concatenating the rows of the target matrix $w_{ij}^{(t)}$ yields and index $k_{ij} = (i-1)c_0 + j$ for the original index pair $(ij)$. Setting

$$m_{(ij)k} := 1, \text{ if } k = k_{ij}, \text{ and } m_{(ij)k} := 0 \text{ otherwise,} \tag{5}$$

results in a feature matrix that equals the identity matrix $\mathbf{M} = \mathbf{I}$, which is obviously perfectly conditioned. $\mathbf{M}\gamma = \mathrm{vec}(\mathbf{W}^{(t)})$ implies then $w_{ij}^{(t)} = \gamma_{k_{ij}}^{(1)}$.

The open question is how this feature matrix can be generated by $\mathbf{W}^{(1)}$ and $\mathbf{W}^{(2)}$. Fig. 1 (c) visualizes the construction. Exactly one path leads from each input neuron $j$ to each output neuron $i$ so that the corresponding $\gamma_{k_{ij}}^{(1)}$ can represent the target weight $w_{ij}^{(t)}$. Accordingly, we define

$$w_{kj}^{(1)} = 1 \text{ and } w_{ki}^{(2)} = 1, \text{ if } k = k_{ij}, \text{ and } w_{kj}^{(1)} = w_{ki}^{(2)} = 0 \text{ otherwise.} \tag{6}$$

$\mathbf{W}^{(1)}$ and $\mathbf{W}^{(2)}$ are maximally sparse, as each feature index $k$ is connected to exactly one input (in $\mathbf{W}^{(1)}$) or one output (in $\mathbf{W}^{(2)}$). Removing such a link would render the feature useless and thus correspond to an effective removal of the feature. For that reason, 1 is a lower bound for row sum and column sum of , respectively. This bound is attained by the above construction. Furthermore, exactly $c_1 = c_0 c_2$ features are provided, which corresponds to the degrees of freedom of a potential target matrix. This number of features is necessary to express any target matrix $\mathbf{W}^{(t)} \in \mathbb{R}^{c_2 \times c_0}$. In conclusion, ID cond features are maximally sparse.

Equivalent dynamics are introduced if the second layer with trainable $\gamma^{(2)}$ assume the typical role of a normalization layer (e.g. BN), while $\gamma^{(2)}$ encodes the linear target layer. Yet, the second layer is

potentially equipped with a nonlinear activation function. This ensures that the loss function in the original parameterization $\mathcal{L}(\mathbf{W}^{(t)})$ is identical to the loss in the parameterization with random features so that $\mathcal{L}(\mathbf{W}^{(t)}) = \mathcal{L}(\gamma^{(1)})$ for each target layer. It follows that also $\nabla\mathcal{L}(\mathbf{W}^{(t)}) = \nabla\mathcal{L}(\gamma^{(1)})$. $\square$

ID cond features that attain the minimum width requirement by utilizing $\gamma^{(2)}$ in addition to $\gamma^{(1)}$ create only one representative intermediary neuron for one of the input neurons (see Fig. 1 (d)). The corresponding weight matrices are defined as:

$w_{kj}^{(1)} = 1$ and $w_{ki}^{(2)} = 1$, if $k = k_{ij}$ for $i > 1$. $w_{k1}^{(1)} = 1$ and $w_{ik}^{(2)} = 1$ if $k = k_{11}$ for $j = 1$ and all $i$.

and $w_{kj}^{(1)} = w_{ki}^{(2)} = 0$ otherwise. Note that we skip the feature indicees $k_{i1}$ for $i > 1$. The corresponding training dynamics change due to the quadratic nature of the parameterization. Yet, solving the student-teacher setup for this parameterization is still straight-forward because of the high sparsity of the features and known solution. In comparison, Ortho cond features, which are generally not sparse, lead to suboptimal reconstruction error in comparison, even though they are also well conditioned.

### A.2.2 PROOF OF THEOREM 4.1

*Statement* (Target Alignment, Thm. 4.1). Assume that $\mathbf{M}$ has rank $r$ and singular value decomposition $\mathbf{M} = \mathbf{USV}^T$ with singular values sorted in decreasing order. Let $\mathbf{P}$ be a permutation of the rows of the target $\mathbf{W}^{(t)}$ and $\mathbf{v}^{(t)}(\mathbf{P}) = \text{vec}(\mathbf{PW}^{(t)})$ be the flattened vector representation of the permuted target. Then, the approximation error is minimized by $\min_{\mathbf{P}} \min_{\gamma} \left\| \mathbf{M}\gamma - \mathbf{v}^{(t)}(\mathbf{P}) \right\|^2 = \min_{\mathbf{P}} \left\| \mathbf{P}_r \mathbf{U}^T \mathbf{v}^{(t)}(\mathbf{P}) \right\|^2$, where $\mathbf{P}_r = \begin{bmatrix} \mathbf{0} & \mathbf{0} \\ \mathbf{0} & \mathbf{I}_r \end{bmatrix}$ projects a vector to its last $r$ components.

*Proof.* In contrast to the previous theorem, we assume that we might not have sufficiently many degrees of freedom to attain perfect reconstruction. Hence, our objective is to minimize the reconstruction error using all our options, including learning the $\gamma^{(1)}$ parameters and permuting the target neurons to align the random features with the target:

$$\min_{\mathbf{P}} \min_{\gamma} \left\| \mathbf{M}\gamma - \mathbf{v}^{(t)}(\mathbf{P}) \right\|^2. \tag{7}$$

To do so, let us first solve the inner optimization problem (i.e. fix the permutation $\mathbf{P}$) and find the optimal parameters given the flattened permuted target $\mathbf{v}^{(t)}(\mathbf{P})$. It is well known that the least square solution is given by $\hat{\gamma} = \mathbf{M}^+ \mathbf{v}^{(t)}(\mathbf{P})$, where $\mathbf{M}^+$ denotes the Moore-Penrose inverse of the feature matrix $\mathbf{M}$. Utilizing the singular value decomposition of $\mathbf{M} = \mathbf{USV}^T$, this leads to the solution

$$\hat{\gamma}(\mathbf{P}) = \mathbf{M}^+ \mathbf{v}^{(t)}(\mathbf{P}) = (\mathbf{M}^T\mathbf{M})^{-1}\mathbf{M}^T\mathbf{v}^{(t)}(\mathbf{P}) = \mathbf{V}(\mathbf{S}^T\mathbf{S})^{-1}\mathbf{S}^T\mathbf{U}^T\mathbf{v}^{(t)}(\mathbf{P}). \tag{8}$$

Plugging this solution into the objective, Eq. (7), turns the nested optimization problem into

$$\min_{\mathbf{P}} \min_{\gamma} \left\| \mathbf{M}\gamma - \mathbf{v}^{(t)}(\mathbf{P}) \right\|^2 = \min_{\mathbf{P}} \left\| \mathbf{M}\hat{\gamma} - \mathbf{v}^{(t)}(\mathbf{P}) \right\|^2$$

$$= \min_{\mathbf{P}} \left\| \left[ \mathbf{USV}^T\mathbf{V}(\mathbf{S}^T\mathbf{S})^{-1}\mathbf{S}^T\mathbf{U}^T - \mathbf{I} \right] \mathbf{v}^{(t)}(\mathbf{P}) \right\|^2$$

$$= \min_{\mathbf{P}} \left\| \left[ \mathbf{S}(\mathbf{S}^T\mathbf{S})^{-1}\mathbf{S}^T - \mathbf{I} \right] \mathbf{U}^T\mathbf{v}^{(t)}(\mathbf{P}) \right\|^2,$$

where we have used that $\mathbf{UU}^T = \mathbf{I}$ (and $\mathbf{V}^T\mathbf{V} = \mathbf{VV}^T = \mathbf{I}$) and the fact that orthogonal transformations $\mathbf{U}$ leave the l2-norm invariant.

It is easy to see that the matrix $\left[ \mathbf{S}(\mathbf{S}^T\mathbf{S})^{-1}\mathbf{S}^T - \mathbf{I} \right]$ defines a projection $\mathbf{P}_r = \begin{bmatrix} \mathbf{0} & \mathbf{0} \\ \mathbf{0} & \mathbf{I}_r \end{bmatrix}$. Note that $\mathbf{S} = \begin{bmatrix} \mathbf{D} \\ \mathbf{0} \end{bmatrix}$, where $\mathbf{D}$ is a diagonal matrix containing the nonzero singular values of $\mathbf{M}$ on the diagonal. Accordingly, we have $\mathbf{S}(\mathbf{S}^T\mathbf{S})^{-1}\mathbf{S}^T = \begin{bmatrix} \mathbf{D} \\ \mathbf{0} \end{bmatrix} \mathbf{D}^{-2} \begin{bmatrix} \mathbf{D} & \mathbf{0} \end{bmatrix} = \begin{bmatrix} \mathbf{DD}^{-2}\mathbf{D} & \mathbf{0} \\ \mathbf{0} & \mathbf{0} \end{bmatrix} = \begin{bmatrix} \mathbf{I} & \mathbf{0} \\ \mathbf{0} & \mathbf{0} \end{bmatrix}$. It thus follows that $\min_{\mathbf{P}} \min_{\gamma} \left\| \mathbf{M}\gamma - \mathbf{v}^{(t)}(\mathbf{P}) \right\|^2 = \min_{\mathbf{P}} \left\| \mathbf{P}_r \mathbf{U}^T\mathbf{v}^{(t)}(\mathbf{P}) \right\|^2 = \min_{\mathbf{P}} \left\| \mathbf{U}_2^T\mathbf{v}^{(t)}(\mathbf{P}) \right\|^2$, where $\mathbf{U} = \begin{bmatrix} \mathbf{U}_1 & \mathbf{U}_2 \end{bmatrix}$. $\square$

This theorem gives us a criterion to evaluate the quality of permutations in projection feature dimensions away that do not align well with the target. We will approximately minimize this criterion in a Greedy algorithm that permutes two rows of the target in each iteration. As the feature matrix $\mathbf{M}$ is generally extremely large, i.e., of the order of $c_0 c_2$ times $c_0 c_2$, computing the singular value decomposition of $\mathbf{M}$ becomes computationally infeasible for realistic neural network architectures. This also limits our ability to precondition arbitrary random feature matrices. In contrast, our random feature proposals induce known singular value decompositions that do not require an explicit numerical estimation. Sparse solutions are particular helpful to construct solutions with low error. The following theorems explore how many features we can expect to require in this setting.

### A.2.3 PROOF OF THEOREM 4.2

*Statement* (Matching sparse targets and features, Thm. 4.2). Let the target matrix $\mathbf{W}^{(t)}$ be a sparse matrix with in-degrees $d_i = \sum_j \delta_{w_{ij}^{(t)} \neq 0}$ and the feature matrix $\mathbf{M}$ correspond to a random $Ber(p)$ matrix (i.e. sparse Id cond features). Then, the probability that the (permuted) target can be accurately matched is upper bounded by $\mathbb{P}(\mathbf{W}^{(t)} \subset \mathbf{M}) \leq \Pi_i \left(1 - \left(1 - p^{d_i}\right)^{c_2}\right)$.

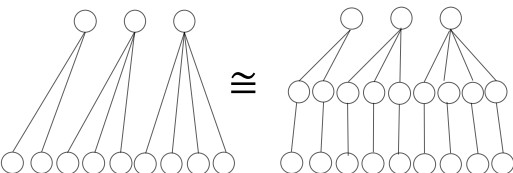

Figure 5: Visualization of construction to obtain the lower bound in Thm. 5. Left: Target network. Right: Random features used to approximate the target.

*Proof.* We consider here the general setting where we assume that a sparse target network is given and we can approximate this target with a random subset of the ID cond features. The latter is equivalent to a single layer with random Erdoes-Renyi mask with iid Bernoulli entries, where each edge is masked (i.e. is dropped/missing as a feature) with probability $1 - p$, while it exists with probability $p$. A target output neuron can be accurately represented by an output neuron of the feature matrix if all its $d_i$ incoming target edges exist also in the feature matrix. This is the case with probability $p^{d_i}$. The question is now how many feature neurons can one choose from to match a specific target neuron. In principle, one could start with one target neuron and try to find the best match among all available $c_2$ feature output neurons. After a match, the next target neuron could be matched with one of the remaining $c_2 - 1$ feature neurons, etc. The last target neuron would only have one neuron available for a match. Accordingly, the probability of finding a match this way would be relatively low. Yet, this Greedy strategy would likely also not be optimal. Considering all $c_2!$ possible matches would lead to more overall matching options for each neuron. Yet, the number of potential matches would be lower than $c_2$ in most cases. We can exploit this fact to construct an upper bound on the matching probability.

In fact, this bound can also be attained in a special case. Consider Fig. 1 (e) and Fig. 5 for this special case. If all target output neurons are connected to mutually exclusive sets of input neurons, then all student outputs are potential matches for a target neuron, which maximizes the probability of a match. The reason is that the target neurons do not overlap. Thus, a match with one specific target neuron does not reduce the probability of another neuron to match with this target. Target neurons therefore do not compete for matches.

In this case, the probability that a given target neuron $i$ can be matched with any of $c_2$ independent output neurons in the feature matrix is given by $1 - (1 - p^{d_i})^{c_2}$. All these matches are independent. Thus, all targets can be thus matched simultaneously with probability $\prod_{i=1}^{c_2}(1 - (1 - p^{d_i})^{c_2})$. $\square$

As discussed in the main paper, despite being a upper bound, this probability is still relatively small, as we have to match a set of $c_2$ output target neurons with $c_2$ output feature matrix neurons. To increase the probability of matches, we have to increase the number of candidates to match with, as proposed by the next theorem.

### A.2.4 PROOF OF THEOREM 4.3

Ideally, we would be able to match every target neuron with one of $kc_2$ available feature matrix output neurons. However, for general targets, the new matches would not be independent of previous target matches. By making the matches independent by partioning the potential output match candidates, the next theorem gives a relatively conservative bound by not allowing to reuse refused matches. This makes it possible to consider relatively general sparse targets. The theorem provides an intuition how this target sparsity $q$ interacts with the output width increase $k$ and the number of available features $p$.

*Statement* (Lower dimensional target, Thm. 4.3). A random target with iid $Ber(q)$ entries and $c_2^{(t)}$ output neurons can be perfectly matched with probability $1 - \delta$ with random ID cond features with expected density $p$ if $c_2 = kc_2^{(t)}$ with $k \geq \frac{\log\left(1-(1-\delta)^{1/c_2^{(t)}}\right)}{\log(1-(1-q(1-p))^{c_0})}$.

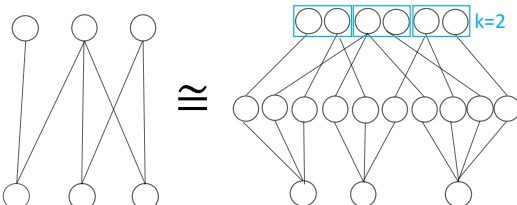

Figure 6: Visualization of construction to obtain the bound on width in Thm. 6. Left: Target network. Right: Random features used to approximate the target. Note that each target neuron can be matched with any output neuron in a set of size $k$.

*Proof.* Fig. 6 visualizes the main idea to partition the number of output candidate matches into disjoint subsets, as this allows for independent matches of target neurons. Accordingly, every target neuron can be matched with one of $k$ candidates. A match happens if all target edges exist also in the feature matrix candidate. A single edge is matched with probability $qp+(1-q)$, since it does not exist in the target with probability $1-q$ making a match obsolete or it exists with probability $qp$ in both the target and the feature matrix. All $c_0$ incoming edges can be matched with probability $(qp+(1-q))^{c_0}$ and a match with at least one of $k$ available candidates is given by $1 - [1 - qp + (1-q))^{c_0}]^k$. The theorem statement follows from requesting that all $c_2$ targets have matches with probability at least $1-\delta$ such that $(1 - [1 - qp + (1-q))^{c_0}]^k)^{c_2} \geq 1 - \delta$. □

Note that, in practice, more than $k$ candidates would be available for matching. In the special case of Thm. 4.2, even $kc_2$ candidates would be available in every matching attempt. However, in general, those matches would not be independent and hence difficult to assess probabilistically.

The previous theorem has still expected a student-teacher setup for matching a single sparse random layer. Deep neural networks offer, however, many more options to represent or approximate different ways to solve a task, thus, providing more options to succeed with sparse networks. Empirical results for random lottery tickets (Liu et al., 2021; Gadhikar et al., 2023b) suggest that random networks with medium sparsity can achieve competitive performance on standard benchmark tasks. This suggests that sparse ID cond features, which are equivalent to such random masks (see Thm. 3.1), can attain the same performance, leading to the following conclusion.

**Conclusion.** *If a task is solvable by training an iid randomly masked neural network with density $p_l$, then $c_1^l = p_l^l c_0^l c_2^l$ random ID cond features can achieve the same generalization performance.*

*Proof.* Explicit lottery ticket existence proofs for random source networks (Gadhikar et al., 2023b) require a substantial overparameterization of the source network in comparison to a target network. For this reason, the theory does not directy support lower width requirements relative to a target network unless the target network has much lower dimensions (and thus we have large $k$). However, we can follow a similar construction idea to obtain a better width requirement by using a 3-layer instead of a 2-layer normalization layer construction to approximate a single sparse random target network. □

This insight concludes our theoretical insights that show that there exist sparse random features, which support lower than previously assumed width requirements for learning only normalization layers.

### A.3   PERMUTATION ALGORITHMS

How can we find a permutation of the rows of the target network or, equivalently, of the feature matrix output neurons and thus the rows of $\mathbf{W}^{(2)}$? It is computationally infeasible to go through all $c_2!$ possible permutations even if we have to evaluate only $\left\|\mathbf{P}_r\mathbf{U}^T\mathbf{v}^{(t)}\right\|^2$ instead of solving each associated linear regression problem utilizing Thm. 4.1. Initially, we have implemented a continuous relaxation of the permutation (Lyu et al., 2020). Yet, this approach was outperformed by just sampling random permutations and picking the best one. This approach would hardly scale to larger output dimensions $c_2$ and also not leverage target information effectively in comparison what we can achieve in Fig. 4.

To obtain Fig. 4, we have implemented a Greedy algorithm that leverages the insight of Thm. 4.1 and matches each output index $i_u$ that corresponds to a set of indicees in the singular value space $u_{(i_uj)k}$ with an output index $i_t$ of the target $w_{i_tj}^{(t)}$. Concretely, we aim to minimize the objective $\min_\pi \sum_{k=c_2c_0-r+1}^{c_2c_0} \left(\sum_{ij} u_{(\pi(i)j)k}w_{t,ij}\right)^2$ searching for a permutation $\pi$ of the row indicees that facilitate the matching. The problem arises from the fact that the sum over $i$ is inside the square. Therefore, sum of the terms $u_{(\pi(i)j)k}w_{t,ij}$ could also have a positive or negative contribution to the sum so that we cannot minimize the terms independently via a pairwise matching. Therefore, we evaluate the change of the score (where the score is $\sum_{k=c_2c_0-r+1}^{c_2c_0} \left(\sum_{ij} u_{(\pi(i)j)k}w_{t,ij}\right)^2$) in response to a match between a row of $\mathbf{U}$ and $\mathbf{W}^{(t)}$. Starting with the indicees $i_u$ of $\mathbf{U}$ with the largest norm, we match them Greedily to the best free row index $i_t$ of the target that minimizes the score. Our specific Python implementation follows and is also shared as part of the supplementary code.

```python
def svd_permute_fc(w01,w02,wt):
    n2, n1, m = w02.size(0), w01.size(1), w01.size(0)
    M = torch.einsum('ik,kj->ijk', w02, w01)
    M = M.reshape((n2*n1,m))
    try:
        U, S, Vh = torch.linalg.svd(M)
        ss = S.size(0)
        Umiss = U[:,ss:]
        Umiss = Umiss.reshape((n2,n1,-1))
        #Greedy
        order_match = torch.argsort(torch.sum(Umiss**2,dim=(1,2)),descending=True)
        perm_Greedy = torch.arange(n2)
        remain = torch.arange(n2)
        for i in order_match:
            i = i.item()
            back = torch.einsum('ijk,ij->k',Umiss,wt[perm_Greedy])
            ind_current = perm_Greedy[i].item()
            score = back-torch.einsum('jk,j->k',Umiss[i,:,:],wt[ind_current,:])
            sc = torch.sum(back**2).item() #torch.zeros(remain.size(0))
            pc = ind_current
            ic = i
            for p in remain:
                p = p.item()
                ip = torch.where(perm_Greedy==p)[0].item()
                x = score+torch.einsum('jk,j->k',Umiss[i,:,:],wt[p,:])..
                ..+torch.einsum('jk,j->k',Umiss[ip,:,:],wt[ind_current,:])
                x = x-torch.einsum('jk,j->k',Umiss[ip,:,:],wt[p,:])
                x = torch.sum(x**2)
                if x.item() < sc:
                    sc = x.item()
```

```
918                         pc = p
919                         ic = ip
920                 perm_Greedy[i] = pc
921                 perm_Greedy[ic] = ind_current
922                 remain = remain[remain!=pc]
923             col_ind = torch.argsort(perm_Greedy)
924             w02 = w02[col_ind]
925             gamma1 =  torch.transpose(Vh,0,1)[:,:ss]/S ..
926             ..@ torch.transpose(U,0,1)[:ss,:] @ wt[perm_Greedy].reshape((-1,))
927             return w02, gamma1
```

If **M** is sparse, a pairwise distance minimization becomes feasible. We can directly maximize the overlap of sparse features with the largest target links with the help of minimum weight matching. In most cases, this sparse matching provides a strong signal for finding a good permutation. We hypothesize that a better matching can often be obtained because the zeros do not incur additional error that propagates through the network as in case of standard random weight matrices that induce dense feature matrices. In addition to the Greedy approach above, we thus also consider a mask overlap maximization, as implemented below.

```
936    def id_permute(w01,w02,wt):
937        n2, n1, m = w02.size(0), w01.size(1), w01.size(0)
938        mask = torch.ones((n2,n2))-torch.einsum('ik,kj->ij', w02, w01)
939        deg = torch.sum(wt**2,dim=1)
940        #Greedy
941        order_match = torch.argsort(deg,descending=True)
942        perm_Greedy = torch.arange(n2)
943        remain = torch.arange(n2)
944        for i in order_match:
945            i = i.item()
946            sc = torch.sum((mask[perm_Greedy,:]*wt)**2).item()
947            ind_current = perm_Greedy[i].item()
948            pc = ind_current
949            ic = i
950            for p in remain:
951                p = p.item()
952                ip = torch.where(perm_Greedy==p)[0].item()
953                x = sc+torch.sum((mask[p,:]*wt[i,:])**2)..
954                ..+torch.sum((mask[ind_current,:]*wt[ip,:])**2)
955                x = x-torch.sum((mask[ind_current,:]*wt[i,:])**2)..
956                ..-torch.sum((mask[p,:]*wt[ip,:])**2)
957                if x.item() < sc:
958                    sc = x.item()
959                    pc = p
960                    ic = ip
961            perm_Greedy[i] = pc
962            perm_Greedy[ic] = ind_current
963            remain = remain[remain!=pc]
964        col_ind = perm_Greedy
965        w02=w02[col_ind]
966        err1 = torch.sqrt(torch.mean((mask[col_ind]*wt)**2)).item()
967        #second approach to identify permutation:
968        mask = torch.einsum('ik,kj->ij', w02, w01)
969        x, _ = torch.sort(torch.abs(wt).flatten(),descending=True)
970        dd=n1*n2
971        p=(m+n2-1)/dd
        thr = min(int(dd*2*p/(1+p)),dd-1)
        thr = x[thr]
        maskt = torch.where(torch.abs(wt)>thr,1.0,0.0)
        dist_mat_rows = torch.cdist(maskt,mask,p=2)
```

```
row_ind, col_ind = linear_sum_assignment(dist_mat_rows.cpu().detach().numpy())
mask = mask[col_ind]
err = torch.sqrt(torch.mean((mask*wt-wt)**2)).item()
if err < err1:
    w02=w02[col_ind]
    err=err1
return err, w02, col_ind
```

