# OpenReview forum: "Random Features for Normalization Layers"
_ICLR.cc/2026/Conference — ICLR 2026 Conference Withdrawn Submission_

### Official Review · Reviewer_DhS5 · 2025-10-27

**Soundness:** 1
**Presentation:** 2
**Contribution:** 1
**Rating:** 0
**Confidence:** 5

**Summary:**

This paper investigates the expressiveness and trainability issues associated with training only normalization layer parameters (like BatchNorm) in neural networks while keeping other weights frozen at random initialization. The authors argue that standard random weight initializations lead to an ill-conditioned learning task for the normalization parameters (justified via Theorem 2.2), citing prior empirical difficulties. They propose alternative "ID cond" and "Ortho cond" random feature constructions that are provably well-conditioned. They show that the ideally conditioned "ID cond" features result in a parameterization equivalent to standard training (Theorem 3.1). The paper also analyzes permutation invariance for reducing feature requirements, finding benefits primarily for sparse targets (Theorems 4.1-4.3). The work aims to provide a theoretical answer to questions raised by Frankle et al. (2021).

**Strengths:**

* Addresses an interesting and relevant open question regarding the expressiveness of normalization layers and random features.

**Weaknesses:**

* **Fatal Flaw** The theoretical justification for the ill-conditioning of standard features (via Theorem 2.2) is invalid. The proof of Theorem 2.2 relies on applying the Marchenko-Pastur law for block-independent matrices (Bryson et al. 2021, Thm 1.3). However, the feature matrix $M$ structure violates the required inter-block independence due to shared $W^{(2)}\_{ik}$ terms for entries $m\_{(iJ)k}$ and $m\_{(iJ')k}$ when $J \neq J'$. I cannot see any way of defining the blocks that will remain faithful to block-wise condition of Thm 1.3 by Bryson.
This invalidates the proof of Theorem 2.2 and removes the paper's core theoretical explanation for why standard features perform poorly.
* **Limited Novelty and Impact of Sound Results:** The theoretically sound contributions (Sec 3, Sec 4) demonstrate limited novelty (assessed as Simple/Trivial) and lead to largely unsurprising conclusions.
* The core result of Section 3 (Thm 3.1) shows that an ideally conditioned construction ($M=I$) using $c_1 = c_0 c_2$ features recovers the original parameterization. This essentially confirms the expected outcome: having degrees of freedom equal to the target dimension allows reconstruction. It offers little insight into parameter reduction or the specific utility of *random* features, as it constructs *ideal* ones.
* The permutation analysis in Section 4 (Thms 4.1-4.3) uses standard techniques to quantify the intuitive idea that permutations improve matching probabilities, especially for sparse targets/features, and that output overparameterization helps. These results, while sound, do not present fundamentally surprising mechanisms or insights.
* The impact of these sound but simple results is further diminished by the fatal flaw in the argument motivating them (the flawed proof of ill-conditioning for standard features).

**Questions:**

* Can Theorem 2.2 be proven without relying on this invalid invocation? Is there an alternative RMT result that applies to this specific dependency structure?

---

### Official Review · Reviewer_u5Ba · 2025-11-01

**Soundness:** 4
**Presentation:** 2
**Contribution:** 3
**Rating:** 2
**Confidence:** 3

**Summary:**

This paper studies the question: what are sufficient conditions on feature matrices such that training normalization normalization layers will produce a good solution. In particular, the authors argue that the conditioning of the feature matrix is the crucial factor and so develop random features that are well-conditioned. They also then show how to adapt the random features to the setting at hand by permuting them to align the dominant directions of the problem with dominant directions.

**Strengths:**

The paper solves an open question posed by [Frankle et al 2021].  I think the development does a nice job of breaking down general random features, then improved conditioning, then adapting to particular setting via permutation. There are interesting results with implications for how we think about model training.

**Weaknesses:**

The paper would benefit substantially from a clear definition of the problem and significantly more precise theorem statements. It would also be helpful to have a justification of the problem (both in the rebuttal and in the paper) — from my perspective, I do not see the why to set the architecture up this way; other papers have done it so that is some justification, but I would like to understand better why.

Once I accept the model:

In my understanding, the paper talks about how the choice of features in M improves the conditioning of the learning problem of the gamma parameters. Thus, there are tradeoffs between how much M is tied to the original learning problem, how much we need to blow up gamma, and accuracy. It would be helpful to have clearer prose describing the tradeoffs in each section. It would also be helpful to talk about the dimensionality of gamma in each result. Also, are there results for accuracy? I think the writing and presentation require significant polishing before publication.

1. What _specifically_ is good about only training batch norm? Does it improve the dynamics / loss landscape? If you’re anyway going to update c0*c2 batch norm parameters, why not just fine-tune the original weight matrix? I see the paragraph in the introduction discussing this but can you specify more what particularly is good about batch norm?
2. Re clearer definition of the problem: it would be helpful to spell out the function that the teacher is computing, the function that the student is computing, exactly where the normalization layers sit relative to the frozen weight matrices. (Also, why are these called normalization weights? Aren't they generic trainable parameters?)
3. Why is this a good / valid / justified architectural model to study this question? I also am not familiar Giannou's and Burkholz's papers and so would appreciate some justification of the model in your paper, as well.
3. Figure 1 is useful but would benefit from clearer labels, depicting the location of the normalization layers.

**Questions:**

In addition to the points raised above, some more concrete questions:

1. How does the loss function in (1) account for permutation invariance? Are the random features permutationally invariant so it doesn’t matter?
2. “Equivalent to learning in the original parameterization” is ill-defined. Do you mean the architectures are equivalent in what they can represent? Or do you mean the dynamics map 1:1? (Thm 3.1, maybe some other places, too). Based on proof, looks like both — theorem statement should include this.
3. Is it true that Thm 3.1 is just saying that if you remove the trained weights and use identity matrix instead, you can train the normalization parameters instead and get equivalent dynamics? I don’t understand how you can replace learned features with random orthogonal features and get the exact same outcome? There has to be some caveat that I’m missing — maybe some particular problem regime, increase in dimensionality of trained vector, longer training time? There must be some clear tradeoff between improving the geometry of the loss function independent of training data and accuracy wrt data. For all of Section 3, is the dimensionality of gamma the same?
4. Theorem 4.1 — I don’t understand the statement. Are you defining the minimizer or talking about the minimum value? Should there be argmin?
5. The idea behind designing M is to make the conditioning good — but somehow there is still benefit to aligning the dominant directions of the problem to the dominant directions of M. Is there a tension here that can be optimized? Perhaps it is okay for M to be a bit worse in terms of conditioning as long as the problem itself also has similar structure?

---

### Official Review · Reviewer_ct4F · 2025-11-02

**Soundness:** 3
**Presentation:** 2
**Contribution:** 2
**Rating:** 2
**Confidence:** 4

**Summary:**

This paper studies random neural networks with trainable parameters in the normalization layers. The authors claim to identify when random initializations are suitable for training these parameters and propose two random feature initializations that are sparse and have condition number 1. They also provide experimental results analyzing the performance of training the batch normalization parameters while varying the condition number of the matrix  $M$ (which would later be defined). The experiments are conducted on CIFAR-10 and ImageNet.

Note that all their results focus on a single linear layer and ignore the biases. Thus, the network with normalization is of the form $W_2 \Gamma W_1$, where $\Gamma$ is allowed to be tuned and the target network is represented bt a matrix $ W^* $. Assume $W_2\in R^{c0\times c1}$, $W_1\in R^{c1\times c2}$ and $W^*\in R^{c0\times c2}$. $\Gamma$ is the a diagonal matrix of dimension $(c_1,c_1)$.

**Strengths:**

The authors first show that if we have two random matrices $ W_1,W_2 $, then the matrix $M$ which is essentially the Khatri–Rao product of $ W_1,W_2 $ has singular values that follow a distribution in which the smallest singular value can become arbitrarily close to zero. This issue can be mitigated if the hidden dimension is increased, meaning $c_1 >> c_0*c_2 $, which would result in a big network.This result is interesting and provides a potential explanation for why training only a subset of the network parameters may be suboptimal.

The authors also explore different properties of the target matrix that could make it easier to reconstruct.

**Weaknesses:**

As a solution of the condition number issue, the authors propose the use of matrices $W_1,W_2$ such that their Khatri-Rao product is equal to the identity matrix.  By previous results in [1] this is equivalent to just a linear layer $W_2\Gamma W_1 = W$, where each entry of $W$ is one of the entries of $\Gamma$. Thus, Theorem 3.1 simply follows for one linear layer. However, I do not think such a claim can be made once the nonlinearities are introduced and we consider multiple layers. In [1] the first layer’s ReLUs are forced to be always “on” by setting the biases sufficiently large; however, when these parameters are learnable, it is unclear whether it $\textit{``is equivalent to learning a neural network in its original parameterization’’}$

Furthermore these networks are not random anymore. Thus, which is the difference from fine-tuning the initial network?

The rest of the results build on cases that the target matrix is lower dimensional or sparse. However, it would be more interesting if the parameters of the random network, rather than those of the target network, were reduced. Such a result has already been obtained in [1], where the random weight matrices are further randomly sparsified.

[1]: Giannou, Angeliki, Shashank Rajput, and Dimitris Papailiopoulos. "The expressive power of tuning only the normalization layers." arXiv preprint arXiv:2302.07937 (2023).

**Questions:**

1. How theorem 3.1 hold if we have actual non-linearities and more than one layers.

2. Theorem 3.1 in the one linear layer case uses the exact reconstruction. But if we know the target network why not just directly fine-tune it?

3. Which is the contribution of section 4? If the target matrices have specific properties, we need to know which is the target matrix to exploit these properties.

---

### Note · Authors · 2025-11-12

I have read and agree with the venue's withdrawal policy on behalf of myself and my co-authors.